

# Excito-repellency of *Myristica fragrans* Houtt. and *Curcuma longa* L. extracts from Southern Thailand against *Aedes aegypti* (L.)

Phuangthip Bhoopong[1], Theeraphap Chareonviriyaphap[2,3] and Chutipong Sukkanon[1,4]

[1] Department of Medical Technology, School of Allied Health Sciences, Walailak University, Nakhon Si Thammarat, Thailand
[2] Department of Entomology, Faculty of Agriculture, Kasetsart University, Bangkok, Thailand
[3] Royal Society of Thailand, Bangkok, Thailand
[4] Research Excellence Center for Innovation and Health Products, Walailak University, Nakhon Si Thammarat, Thailand

Corresponding author
Chutipong Sukkanon,
chutipong.su@wu.ac.th

## ABSTRACT

The development of local plant extracts as a mosquito repellent is environmentally safe, inexpensive, and easily accessible for residents. In this study, three extracts from two local plants, *Myristica fragrans* Houtt. (flesh and mace) and *Curcuma longa* L. (rhizome) from southern Thailand, were investigated for their inherent repellent activity using the excito-repellency (ER) assay system against insectary-colonized *Aedes aegypti* (L.) (Diptera: Culicidae). The escape responses of mosquitoes exposed to concentrations of 0.5% to 5.0% (w/v) were measured to determine the contact irritant and non-contact repellent properties of each extract. Both the flesh and mace extracts of *M. fragrans* had relatively limited contact irritants (28.1% and 34.6% escape) and non-contact repellent (16.7% and 18.3% escape) activities against *Ae. aegypti*, respectively. The *C. longa* rhizome extract produced higher escape responses in the non-contact (42.6% escape) and contact (41.4% escape) trials at concentrations of 5.0% and 1.0%, respectively. GC-MS analysis found diethyl malate (56.5%) and elemicin (11.7%) to be the main components of the flesh and mace extracts, respectively, while ar-turmerone (24.6%), β-turmerone (15.2%), α-turmerone (10.5%) were the primary constituents of the rhizome extract. Overall, our results indicate that both *M. fragrans* extracts primarily caused *Ae. aegypti* escape through contact irritation. For *C. longa*, lower concentrations (0.5% and 1.0%) exhibited contact irritancy, but higher concentrations (2.5% and 5.0%) exhibited non-contact repellency against *Ae. aegypti*. Although they had limited efficacy, further experiments (e.g., mixing with other plant-based compounds) could enhance the ER of both local plant extracts. Additional evaluation of these extracts against other mosquito species and the ER of their chemical components, either alone or in combination, would also be beneficial for the development of green repellents. Our findings emphasize the possibility of utilizing plant-based mosquito repellent as an alternative personal protection method for future mosquito control programs.

## INTRODUCTION

*Aedes aegypti* (L.) (Diptera: Culicidae) is one of the most notorious, anthropophilic mosquitoes in the world, and is capable of transmitting several tropical infectious diseases, such as dengue fever, chikungunya, yellow fever, and Zika (*Phumee et al., 2019*). Although the majority of patients infected with Zika virus are asymptomatic or develop a mild disease (*Lessler et al., 2016*). However, studies show that infection in early pregnancy can lead to sequelae, such as microcephaly (*Wongsurawat et al., 2018*). In Thailand, the Zika virus (ZIKV) is widespread and has been circulating at a low but sustained level since 2002 (*Ruchusatsawat et al., 2019*). Last year, the ZIKV was predominantly found in the southern provinces of Thailand, where approximately 170 cases were reported in 2020 and 13 cases of congenital microcephaly syndrome were reported since 2016 (*DVBD, 2020*). To date, vaccines against the ZIKV remain under development and specific antiviral therapeutics are lacking. Thus, vector management is the main approach in ZIKV control (*Sharma & Lal, 2017*).

Mosquito repellent is one of the most effective personal protection tools against the ZIKV vector (*Nerio, Olivero-Verbel & Stashenko, 2010*; *Debboun & Strickman, 2013*). In addition to synthetic repellents, such as *N,N*-diethyl-*meta*-toluamide (DEET), picaridin, and IR3535 (*Leal, 2014*), plant-based compounds have received attention over the years as an alternative repellent to reduce human-mosquito contact (*Maia & Moore, 2011*; *Tisgratog et al., 2016*). Both crude extracts and essential oils are recognized for their multiple advantages, including their promising efficacy, eco-friendly properties, minimal adverse effects, inexpensive cost, availability, and local acceptance (*Nerio, Olivero-Verbel & Stashenko, 2010*). Many rural communities still traditionally use several local plants as insect repellents (*Maia & Moore, 2011*); thus, testing and developing novel repellents from these plant extracts could be extremely beneficial in combating the ZIKV vector.

*Myristica fragrans* Houtt. (Magnoliales: Myristicaceae), commonly known as the 'nutmeg tree', is an evergreen spice tree native to the Moluccas Islands of Indonesia, but widely distributed in tropical Asia, including India, Singapore, Malaysia, and Thailand (*Kuete, 2017*). There are two important parts of the *M. fragrans* fruit: the shelled seed (flesh) and the red aril (mace) that surrounds the seed. With a unique fragrance and a slightly sweet, warm taste, both parts are used as spices to flavor baked goods (*Periasamy et al., 2016*). Traditionally, *M. fragrans* is also used to cure disease, such as muscle and joint pain, headaches, diarrhea, and fever (*Kuete, 2017*). In Thailand, the nutmeg tree is called '*Chan-Thet*', and the flesh and mace parts are called '*Luk-Chan*' and '*Rok-Chan*', respectively (*Lim, 2012*). In the literature, the phytochemical properties of *M. fragrans* have been identified, which include the phenylpropanoids, lignans, diphenylalkanes, neolignans, and terpenoids. These compounds exhibit a wide variety of pharmacological effects, including anti-inflammatory, antioxidant, antibacterial, anti-diabetic, and anti-cancer activities (*Ha et al., 2020*). Additionally, *M. fragrans* extract is used as an insecticide and repellant (*Lim,*

*2012*). For mosquitoes, the seed oil of *M. fragrans* effectively kills *Ae. aegypti* ($LC_{50}$ = 28.2 ppm) and *Anopheles stephensi* (Liston) ($LC_{50}$ = 2.22 ppm) larvae (*Senthilkumar, Varma & Gurusubramanian, 2009*; *da Rocha Voris et al., 2018*). *M. fragrans* leaf oil has a high repellency (5–8 h bite protection) against *Aedes albopictus* (Skuse), *Anopheles dirus* Peyton & Harrison, and *Culex quinquefasciatus* Say, but limited repellency against *Ae. aegypti* (1.83 h) (*Tawatsin et al., 2006a*). Interestingly, when *M. fragrans* leaf extract was used as a reducing and stabilizing agent, zinc oxide nanoparticles were extremely effective at killing *Ae. aegypti* first instar larvae ($LC_{50}$ = 3.44 ppm) and adults ($LC_{50}$ = 15.004 ppm) (*Ashokan et al., 2017*).

*Curcuma longa* L. (Zingiberales: Zingiberaceae), commonly known as 'turmeric', is a rhizome and herbaceous perennial plant. Although native to tropical South Asia, especially India, *C. longa* has been extensively cultivated in many countries, such as China, Brazil, and Thailand, where it is called '*Khamin*' (*Das, 2016*; *Lim, 2016*). The rhizome is commonly used as a culinary spice in Asian cuisine, as well as a natural dye (*Das, 2016*). *C. longa* has also been traditionally used for centuries to treat a variety of symptoms, such as asthma, allergy, and diabetic wounds (*Araújo & Leon, 2001*). Numerous phytochemicals have been found in the rhizomes, including curcuminoids, diarylheptanoids, diyrenphenate, isoflavone, monoterpenes, sesquiterpenes, diterpenes, and triterpenoids (*Ayati et al., 2019*). The yellowish active constituent of *C. longa* extract and its essential oil display a variety of remarkable bioactive activities, including antioxidant, anti-neoplastic, antimicrobial, anti-inflammatory, anti-coagulant, and anti-venom properties (*Lim, 2016*). Among these properties are insecticidal and insect repellent properties (*Bengmark, Mesa & Gil, 2009*). For instance, *C. longa* oil (10 mg) has been shown to effectively control *Ae. aegypti*, *Ae. albopictus* and *Culex pipiens* L. larvae with more than 84% mortality in 24 h (*Zhu et al., 2008*). The ethanolic extract of the rhizome and two essential oils (leaf and rhizome; 10 mg/cm$^2$) showed a similar bite-deterrent activity against *Ae. aegypti* when compared to the synthetic repellent DEET (25 nmol/cm$^2$) (*Ali, Wang & Khan, 2015*). In addition, the combination of *C. longa* oil with *Cananga odorata* (ylang-ylang) oil provided more than 100 min of protection against *Ae. aegypti*, *Anopheles minimus* Theobald, and *Cx. quinquefasciatus* (*Phasomkusolsil & Soonwera, 2010*). Turmeric oil (10%) also prevented complete mosquito landing on a human volunteer legs for up to 9 h in the field (*Tawatsin et al., 2006b*).

Before novel plant-derived repellents can be formulated, their efficacy must be quantified in detail (*Tisgratog et al., 2016*). Repellents can be classified as non-contact 'spatial' repellents or contact 'excitation' irritants, based on how the mosquito reacts to the repellent (*Deletre et al., 2016*). Non-contact repellents prevent mosquitoes from moving into a given space by creating an odor barrier (*Achee et al., 2012*). Conversely, contact irritants destroy the physical association between mosquitoes and humans after direct tarsal contact (*Grieco et al., 2007*). Responses derived from some combination of both contact irritancy and non-contact repellency are commonly known as 'excito-repellency (ER)' (*Miller et al., 2009*). The characterization of such ER properties in both synthetic and natural-based active ingredients is key for the further development of repellent formulations (*Grieco et al., 2007*). The hypothesis of this study is that the crude extracts of *M. fragrans* and *C. longa*

must possess either solely or a combination of non-contact repellent and contact-irritant activities.

To test for ER activity, an ideal experimental arena should prevent (repellent testing) and allow (irritant testing) contact between the mosquito and the test substance in the absence of the host (*Deletre et al., 2016*). Fortunately, a bioassay called the 'excito-repellency assay system' was developed to distinguish the ER properties of any compounds by measuring the escape response of mosquitoes exposed to treated surfaces (*Chareonviriyaphap et al., 1997*; *Roberts et al., 1997*; *Sukkanon et al., 2022*). In Thailand, the ER assay system has been used for plant-based compounds since 2008 (*Polsomboon et al., 2008*) to reveal the inherent ER activity of several plants (*Tisgratog et al., 2016*). For example, our previous study showed that the crude extract of *Andrographis paniculata* exhibits a strong non-contact repellency against day-biting *Ae. albopictus* and *Ae. aegypti* at high concentrations (2.5%–5.0% w/v; *Sukkanon et al., 2020*). Contrarily, contact irritancy and non-contact repellency are the primary and secondary actions, respectively, of *Cananga odorata* oil and are mainly used against the night-biting mosquitoes *An. minimus* and *Cx. quinquefasciatus* (*Sukkanon et al., 2022*). Evidently, the ER assay system is a suitable bioassay for differentiating repellency from irritancy. To the best of our knowledge, no study has tested the efficacy of *M. fragrans* and *C. longa* against endemic *Ae. aegypti* using the ER assay system in Southeast Asia, including Thailand. Thus, this study aimed to investigate the differentiation between the contact irritancy and non-contact repellency of selected crude extracts from *M. fragrans* (flesh and mace) and *C. longa* (rhizome) using the ER assay system against colonized-female *Ae. aegypti*, a ZIKV vector in southern Thailand.

## MATERIALS & METHODS

### *Ae. aegypti* (L.)

Filter paper containing dried *Ae. aegypti* eggs were provided by the United States Department of Agriculture (USDA) Center for Medical, Agricultural, and Veterinary Entomology, Gainesville, Florida, USA, hereby called the 'USDA strain'. This laboratory colony has been continuously maintained in the climate-controlled insectarium ($25 \pm 2\,°C$, $80 \pm 10\%$ relative humidity, and 12:12 (Light:Dark) photoperiod) at the Department of Entomology, Faculty of Agriculture, Kasetsart University (KU), Bangkok, Thailand, for more than 20 years. *Aedes aegypti* were reared following the KU standard methods, with slight modifications (*Thanispong et al., 2009*). Briefly, the eggs were hatched in filter water, and the larvae were reared at a density of ~150–200 larvae/tray ($22 \times 33 \times 6$ cm). Larval feed (TetraMin® Tropical Fish Food Flakes, Tetra, Virginia, USA) was provided daily (10 g per tray) until pupation. Pupae were removed daily and allowed to emerge in a screen mesh cage ($30 \times 30 \times 30$ cm). A sugar solution (10% w/v) was provided by dipping a cotton stick in a small, solution-filled glass bottle. No artificial membrane feeding was performed. All mosquitoes were reared from existing eggs and maintained in the KU insectary throughout the experiment. This protocol was approved by Walailak University Animal Ethics Committee (Reference No. WU-ACUC-64034).

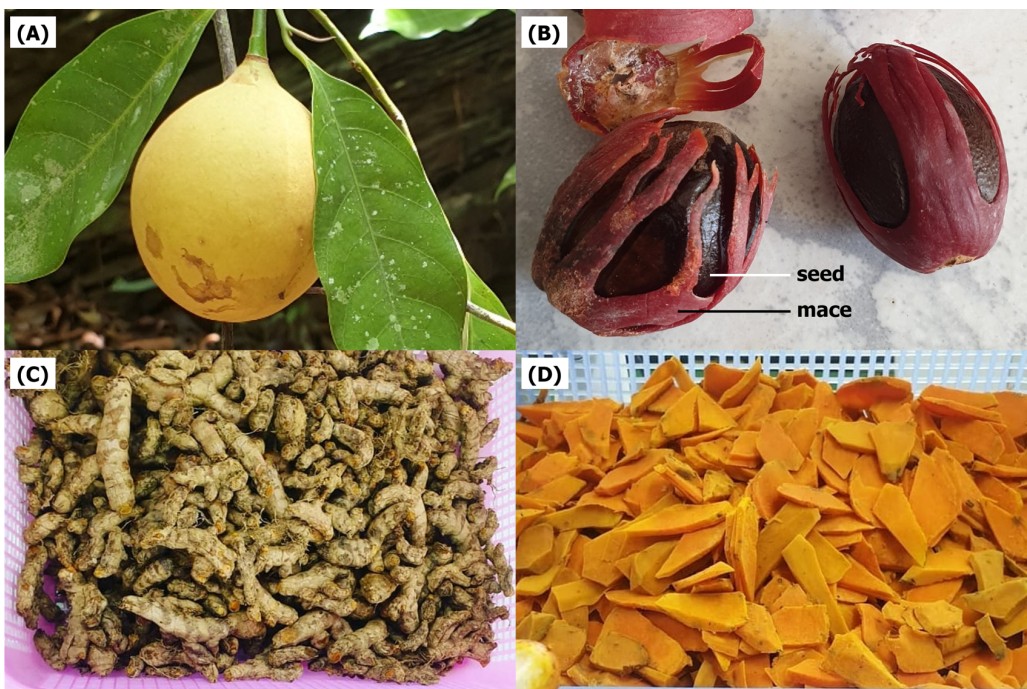

**Figure 1** The fruit (A) and mace (B) of *Myristica fragrans* and the rhizome (C, D) of *Curcuma longa* used for extraction.

## Plant material and extract preparation

A total of three extracts from two local plants were used in the present study. *Myristica fragrans* fruits (Fig. 1A) and *C. longa* rhizomes (Fig. 1C) were obtained in April 2020 from cultivated areas in Nakhon Si Thammarat, Thailand. Both plants were authenticated at the Walailak Botanic Park, Walailak University, Thailand, where they were stored under the voucher specimen numbers 015333 and 01539, respectively. The crude extracts were prepared according to a previous study with minor modification (*Bhoopong et al., 2019*). The flesh (mesocarp) and mace from *M. fragrans* (Fig. 1B) and the rhizomes from *C. longa* (Fig. 1D) were separated, washed, and dried in a hot-air oven at 60 °C. Next, the dried material was powdered using a laboratory blender. *M. fragrans* flesh and mace were extracted in ethanol, while *C. longa* rhizomes were extracted in methanol by adding 600 mL of each respective solvent to 300 g of the given powder. These were macerated for 72 h at 25 °C and 180 rpm in a shaking incubator, then filtered through Whatman No. 1 filter paper (Cytiva, Buckinghamshire, UK). The filtrate was concentrated using a rotary evaporator at 60 °C and reduced pressure. The extracts obtained were stored under refrigeration at 0–4 °C until ready for use (Fig. 2A).

## Paper impregnation

To standardize the ER assay system protocol and allow for finding comparison (*Sukkanon et al., 2022*), each extract solution was prepared with absolute ethanol for *M. fragrans* and methanol for *C. longa* to obtain 0.5%, 1.0%, 2.5%, and 5.0% (w/v). The extract solution

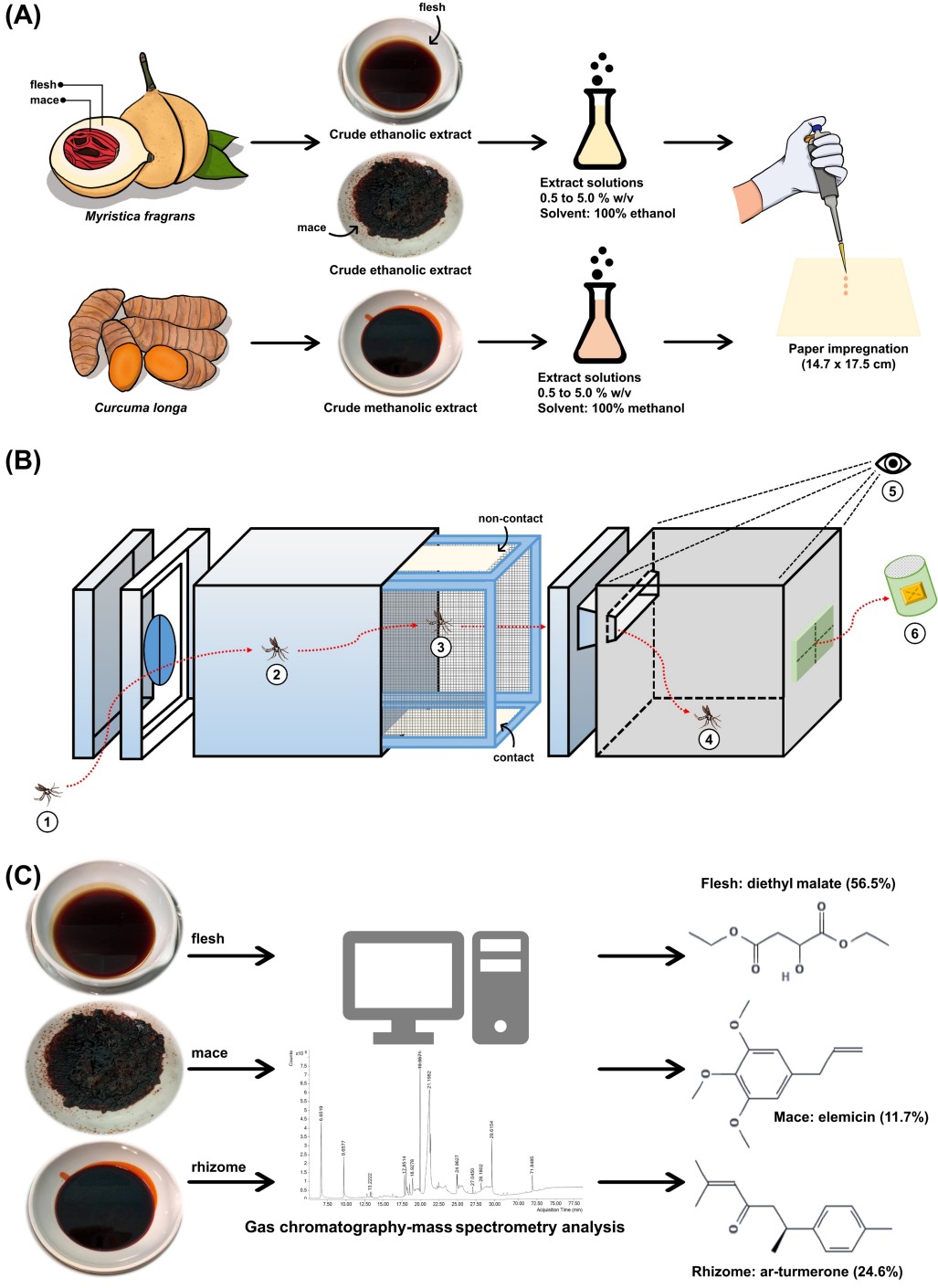

**Figure 2 Study schematic.** (A) The crude ethanolic extract of *Myristica fragrans* (flesh and mace) and crude methanolic extract of *Curcuma longa* (rhizome) were prepared with solvent to concentrations ranging from 0.5% to 5.0% (w/v) and impregnated onto filter papers. (B) The components of the excito-repellency assay system, briefly: (1) 15 female mosquitoes were introduced through a rubber latex door; (2, 3) mosquitoes were allowed 3 min to acclimate inside the metal 

**Figure 2 (…continued)**
screen inner chamber; (3) mosquitoes were exposed to treated paper either with filter papers placed in the inner chamber (contact trial) or with filter papers placed behind a mesh screen (non-contact trial) to control whether direct tarsal contact was possible; (4) mosquitoes could escape *via* an exit portal to the receiving box; (5) experimenter recorded the number of escapes at 1 min interval using the naked eye; and (6) escaped mosquitoes were collected into plastic holding cups. (C) GC-MS analysis was performed on each extract to identify chemical components.

(2.8 mL) was then impregnated on Whatman No. 1 filter papers (14.7 ×17.5 cm) using a calibrated micropipette (Fig. 2A). The solvent was used for control papers. All impregnated papers were placed on aluminum foil and allowed to air dry at room temperature for 30-45 min before use. Papers were prepared and used only once. The test papers contained approximately 183.5, 367, 917.5, and 1,835 $mg/m^2$ of the extract solution for the 0.5%, 1.0%, 2.5%, and 5.0% concentrations, respectively.

## Excito-repellency assay system

The system was designed to test the ER properties of the extracts. When exposed, if the tested compound possesses ER properties, then the mosquito will exhibit an avoidance response by moving away from the compound source. In this experiment, the mosquito would then escape from the treatment chamber into the interconnected receiving chamber (compound-free box; Fig. 2B). The system can measure this escape response as a primary outcome to evaluate the ER properties, where a high percentage of escape indicates high ER properties. There are two designs in the system: non-contact and contact trials. The non-contact trial evaluates repellency by placing the filter paper behind a screen mesh barrier, thus exposing the mosquitoes to the airborne compound molecules inside the chamber and ensuring that the mosquitoes cannot make direct tarsal contact with the treated-filter paper. In the contact trial, the mosquitoes are allowed free physical contact with the treated paper, thus measuring contact excitation or irritancy (Fig. 3). Each treatment chamber was paired with a matched control chamber. Thus, the entire ER assay system consists of two trials of four chambers: (1) non-contact control and treatment chambers; and (2) contact control and treatment chambers. The ER system also measures the knockdown and mortality of escaped mosquitoes and those that remain after 30 min of exposure time (non-escaped mosquitoes). The treatment chambers were made of window-less stainless steel to prevent external light, thus making observation inside the chamber impossible. The top of the receiving chamber was made of transparent plastic sheets to enable escape observation.

Prior to testing, non-blood-fed 4 to 8 day old females were deprived of the sugar meal for 24 h (water only). Then, 15 active female mosquitoes were gently transferred into each of the four chambers using a mouth aspirator, after which the chamber was closed. The test mosquitoes were allowed to acclimate for 3 min inside the chamber. To start the testing, the exit portal slot of each chamber was subsequently opened and the timer started. The number of escaped mosquitoes were counted at 1 min intervals for a period of 30 min with the naked eye. Escaped mosquitoes were removed from the receiving chamber and transferred to holding cups containing the sugar meal (Fig. 2B). At 30 min post-exposure,

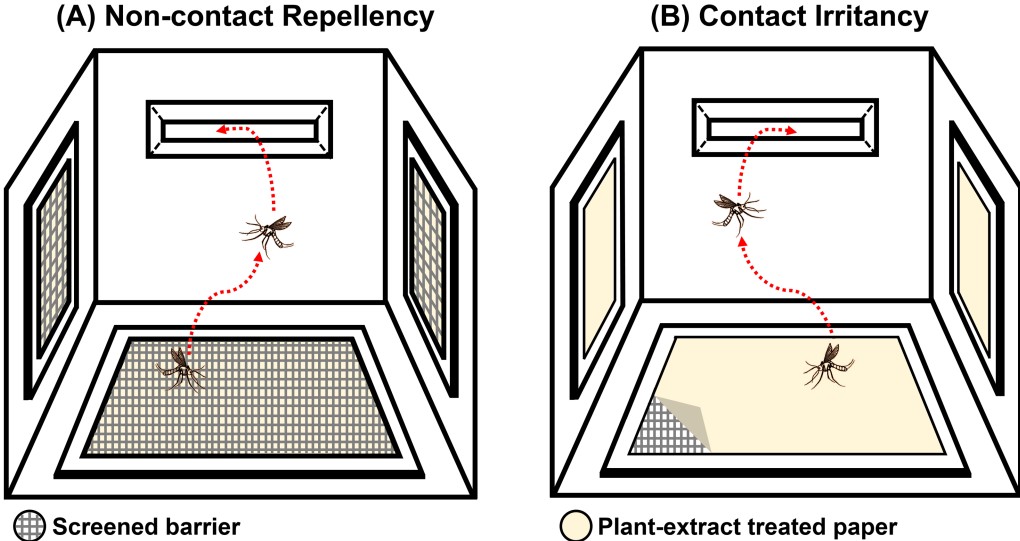

**(A) Non-contact Repellency** **(B) Contact Irritancy**

⊞ **Screened barrier** ○ **Plant-extract treated paper**

**Figure 3** **Illustration of the (A) non-contact and (B) contact chambers of excito-repellency assay system configurations.** The mesh screen in the non-contact chamber prevents mosquitoes from having direct tarsal contact with the plant-extract treated papers, whereas, the mosquito is allowed physical contact with the treated paper by placing it in the inner contact chamber, respectively.

the non-escaped mosquitoes were also removed from each treatment chamber and were transferred to separate holding cups. The number of knockdowns were immediately recorded. All mosquitoes were then held under optimal conditions for 24 h to observe for mortality. Four replications were performed. After each replication was terminated, a new group of 15 female mosquitoes per ER chamber were used for the new replication (60 mosquitoes/replica). The experimental room was set at 25 ± 2 °C and 80 ± 10% relative humidity. All tests were performed between 08:00 to 16:30 h.

## Gas chromatography-mass spectrometry (GC-MS) analysis

GC-MS (Agilent Technologies, USA) was performed following a previous study with modifications (*Matulyte et al., 2019*). Prior to GC-MS, 20 mg of each extract was dissolved in 1 mL of an extract solvent, then centrifuged at 4 °C at 10,000 rpm for 5 min and filtered through a 0.2 µm nylon membrane filter. Analysis was then performed on a gas chromatograph (GC-7890B) and a mass selective detector (MSD-5977B) system. The column thickness was 30 m × 250 µm × 0.25 µm (Agilent 19091S-433). The flow rate of the helium carrier gas was set at 1 mL/min. The oven temperature was maintained at 40 °C for 5 min after injection, then programmed at 5 °C per min, followed by 300 °C for 5 min. The split ratio was 1:10. The mass detector electron ionization was 70 eV. Potential compounds were identified using the Wiley10 and NIST14 libraries, and the match score criteria were accepted at ≥90% (Fig. 2C).

## Statistical analysis

All data were recorded, edited, and entered using GraphPad Prism (GraphPad Software, San Diego, CA, USA). The percentage of escaped mosquitoes was calculated as follows:

(number of escaped mosquitoes / total number of tested mosquitoes) × 100. The mean percent escape and standard error of the mean (SEM) were then determined. The mean percent escape at various concentrations was compared statistically using the Kruskal-Wallis test followed by Dunn's Post Hoc Multiple Comparison test with a significance level of $P < 0.05$. The Mann–Whitney $U$ test was used to compare the mean percent escape between non-contact and contact trials at the same concentrations. Abbott's formula (*Abbott, 1925*) was applied to adjust for the percentage of knockdown at 30 min post-exposure, the percentage of mortality at 24 h post-exposure, and the mean percent escape if the outcome in matched control chambers was 5%–20%. The statistical analysis of the ER assay system has been previously described (*Roberts et al., 1997*). Briefly, using initial escape data at 1 min intervals, Kaplan–Meier survival analysis was used to estimate mosquito escape rates and compare differences in mosquito response. Non-escaped and escaped mosquitoes were treated as 'survived' and 'death', respectively, for computational purposes (*Roberts et al., 1997*). A log-rank method (*Mantel & Haenszel, 1959*) was then applied to compare differences in escape patterns between control and treatment test formats, non-contact and contact trials, and between extract concentrations. A set at 95% confidence interval ($P < 0.05$) was considered statistically significant.

## RESULTS

The escape response of mosquitoes exposed to the 0.5%, 1.0%, 2.5%, and 5.0% extract concentrations and the mean percentage of escape from the non-contact and contact trials is presented in Table 1. No significant difference ($P > 0.05$) in escape percentages was found between any control group in either trial, regardless of concentration (Table 1). Higher escape percentages were found in both treatments of all three extracts compared to the control groups, except for the non-contact trial of 1.0% *M. frangrans* flesh extract. The flesh extract elicited the highest escape response in the non-contact (16.67% escape) and contact (34.55% escape) trials at the 5.0% and 2.5% concentrations, respectively, after adjustment using Abbott's formula. The non-contact trials of the flesh extract produced no significant differences in mean escape percentage, regardless of concentration, but the 0.5% flesh extract yielded a significantly lower mean escape percentage than the 2.5% concentrations ($P = 0.0253$) in the contact trial (Table 1). The 5.0% *M. frangrans* mace extract produced the highest escape response in both the contact (28.07% escape) and non-contact (18.33% escape) trials; however, it produced no significant differences in the mean escape percentage during the contact trials, regardless of concentration. In the non-contact trial, the 0.5% mace extract caused a significantly lower mean escape percentage compared to the 5.0% extract (Table 1). For the rhizome extract, the highest escape responses were observed in the non-contact (42.60% escape) and contact (41.38% escape) trials at the 5.0% and 1.0% concentrations, respectively. No significant differences in the mean escape percentages were observed in either the contact or non-contact trials (Table 1). When the non-contact and contact trials were compared, the 1.0% and 2.5% flesh extract exhibited a significantly higher escape percentage in the contact trial than in the non-contact trial ($P = 0.0286$). For the mace extract, the only significant difference was observed at a concentration of

**Table 1  Mean percent escape (±SEM) of female *Ae. aegypti* exposed to *Myristica fragrans* and *Curcuma longa* extracts using ER assay system.**

| Extracts | ER assay | Conc. | | | | | |
|---|---|---|---|---|---|---|---|
| | | (% w/v) | N | Control[*] | N | Treatment[**] | % Corrected[***] |
| *M. fragrans* (flesh) | Non-contact | 0.5 | 60 | $1.67 \pm 1.67$ | 60 | $8.33 \pm 1.67^a$ | |
| | | 1.0 | 60 | $8.33 \pm 1.67$ | 60 | $8.33 \pm 1.67^a$ | 0.00 |
| | | 2.5 | 60 | $6.67 \pm 2.72$ | 60 | $13.33 \pm 2.72^a$ | 7.14 |
| | | 5.0 | 60 | $0.00 \pm 0.00$ | 60 | $16.67 \pm 4.30^a$ | |
| | Contact | 0.5 | 60 | $5.00 \pm 1.67$ | 60 | $15.00 \pm 1.67^a$ | 10.53 |
| | | 1.0 | 60 | $6.67 \pm 2.72$ | 60 | $35.00 \pm 1.67^{ab}$ | 30.35 |
| | | 2.5 | 60 | $8.33 \pm 1.67$ | 60 | $40.00 \pm 2.72^b$ | 34.55 |
| | | 5.0 | 60 | $3.33 \pm 1.92$ | 60 | $28.33 \pm 7.39^{ab}$ | |
| *M. fragrans* (mace) | Non-contact | 0.5 | 60 | $1.67 \pm 1.67$ | 60 | $3.33 \pm 1.92^a$ | |
| | | 1.0 | 60 | $5.00 \pm 3.19$ | 60 | $13.33 \pm 2.72^{ab}$ | 8.42 |
| | | 2.5 | 60 | $5.00 \pm 1.67$ | 60 | $8.33 \pm 3.19^{ab}$ | 3.51 |
| | | 5.0 | 60 | $1.67 \pm 1.67$ | 60 | $18.33 \pm 3.19^b$ | |
| | Contact | 0.5 | 60 | $1.67 \pm 1.67$ | 60 | $20.00 \pm 2.72^a$ | |
| | | 1.0 | 60 | $1.67 \pm 1.67$ | 60 | $16.67 \pm 1.92^a$ | |
| | | 2.5 | 60 | $3.33 \pm 1.92$ | 60 | $20.00 \pm 2.27^a$ | |
| | | 5.0 | 60 | $5.00 \pm 1.67$ | 60 | $31.67 \pm 5.69^a$ | 28.07 |
| *C. longa* (rhizome) | Non-contact | 0.5 | 60 | $5.00 \pm 5.00$ | 60 | $15.00 \pm 6.87^a$ | 10.53 |
| | | 1.0 | 60 | $3.33 \pm 1.92$ | 62 | $9.79 \pm 3.40^a$ | |
| | | 2.5 | 60 | $6.67 \pm 2.72$ | 60 | $31.67 \pm 7.39^a$ | 26.79 |
| | | 5.0 | 61 | $4.90 \pm 1.63$ | 61 | $42.60 \pm 9.57^a$ | |
| | Contact | 0.5 | 61 | $1.56 \pm 1.56$ | 61 | $25.42 \pm 9.21^a$ | |
| | | 1.0 | 60 | $3.33 \pm 3.33$ | 61 | $41.38 \pm 9.14^a$ | |
| | | 2.5 | 61 | $3.33 \pm 1.92$ | 60 | $18.33 \pm 5.69^a$ | |
| | | 5.0 | 61 | $3.33 \pm 1.92$ | 61 | $22.29 \pm 14.03^a$ | |

**Notes.**

N, number of mosquitoes.

[*]Mean percent escape in control group showed no significantly difference ($P > 0.05$) between concentrations in the same ER trial using Dunn's multiple comparisons test.

[**]Different letter indicates significant differences ($P < 0.05$) in mean percent escape between concentrations in the same treatment group (non-contact or contact) of the same extract using Dunn's multiple comparisons test.

[***]Percent escape adjusted with paired controls using Abbott's formula.

% w/v means percent of weight (g) of plant extract in the total volume of solution.

0.5%, indicating a greater escape percentage in the contact than in the non-contact trial ($P = 0.0225$). No significant difference in escape response between the trials was found for the rhizome extract, regardless of extract concentration. No knockdown or mortality was observed during the 30 min exposure period or 24 h post-exposure for any extract or concentration.

Survival curves were generated to show the mean proportion of mosquitoes remaining in the ER chambers at 1 min intervals during the 30 min exposure time using raw escape data (Fig. 4). In the non-contact trial, the flesh extract produced similar escape patterns regardless of its concentration, but with a quicker escape rate observed at 5.0% (Fig. 4A). In the contact trial, 0.5% flesh extract elicited fewer escaped mosquitoes with a significantly

different escape pattern compared to the higher concentrations, except for the 5.0% concentration (Fig. 4B). In the non-contact trial, the 1.0% and 5.0% mace extracts had a significantly greater escape pattern than the 0.5% extract (Fig. 4C). Although the 5.0% mace extract yielded the highest escape rate in the contact trial, the escape patterns were not significantly different at any concentration (Fig. 4D). In the non-contact trial, the 0.5% and 1.0% rhizome extract exhibited similar escape patterns. Furthermore, the 2.5% and 5.0% concentrations resulted in a more prominent and rapid escape rate (Fig. 4E). In the contact trial, *Ae. aegypti* exhibited a higher escape rate with a faster exit time only at the 1.0% concentration (Fig. 4F).

Multiple paired comparisons of the escape patterns between the non-contact and contact trials were made for each extract type and concentration using log-rank analysis (Table 2). Significant differences were seen for the *M. fragrans* flesh extract at 1.0% and 2.5%, and for the mace extract at 0.5%. For the rhizome extract, the escape patterns were significantly different at the 1.0% and 5.0% concentrations. Overall, the contact trial had a greater escape rate than the non-contact trial for all significant differences seen, except for the 5.0% rhizome extract.

The pattern of escape in the non-contact and contact trials was also compared between concentrations of the same extract (Table 3). Only two pairwise comparisons (0.5%–1.0% and 0.5%–2.5%) revealed significant differences in the contact trial using the flesh extract; however, there were no significant differences in the non-contact group (Table 3, Figs. 4A–4B). In contrast, significant differences in the mace extract were only seen in the 0.5%–1.0% and 0.5%–5.0% comparisons of the non-contact trials (Table 3, Figs. 4C–4D). There were no significant differences found between the two lowest concentrations (0.5%–1.0%) and the two highest concentrations (2.5%–5.0%) of the rhizome extract in the non-contact trials (Table 3, Fig. 4E). Only two pairwise comparisons (1.0%–2.5% and 1.0%–5.0%) were significantly different in the contact trials, since 1.0% of the rhizome extract triggered the highest escape rate (Table 3, Fig. 4F). Overall, these findings indicate that contact irritancy primarily contributes to the escape response of *Ae. aegypti* for both *M. fragrans* extracts. The *C. longa* extract exhibited contact irritancy at lower concentrations (0.5% and 1.0%) but demonstrated non-contact repellency at higher concentrations (2.5% and 5.0%).

The GC-MS results on the 20 mg extract sample soaked in 1 mL of solvent showed a total of 23, 51, and 51 organic compounds in the flesh, mace, and rhizome extracts, respectively. Table 4 shows the primary phytochemical constituents of each extract. Diethyl malate was the main compound of the *M. fragrans* flesh extract, comprising more than half (56.45%) of all chemical components. The mace extract contained twice as many compounds as the flesh extract, with elemicin (11.68%), a phenylpropanoid, as the main component. Three sesquiterpenes were the main component of the *C. longa* rhizome extract: ar-turmerone (24.56%), β-turmerone (15.24%), and α-turmerone (10.52%).

## DISCUSSION

Overall, our findings indicate that the *M. fragrans* flesh and mace extracts have limited ER action against *Ae. aegypti* at concentrations of 0.5%–5.0% (w/v). The *C. longa* rhizome

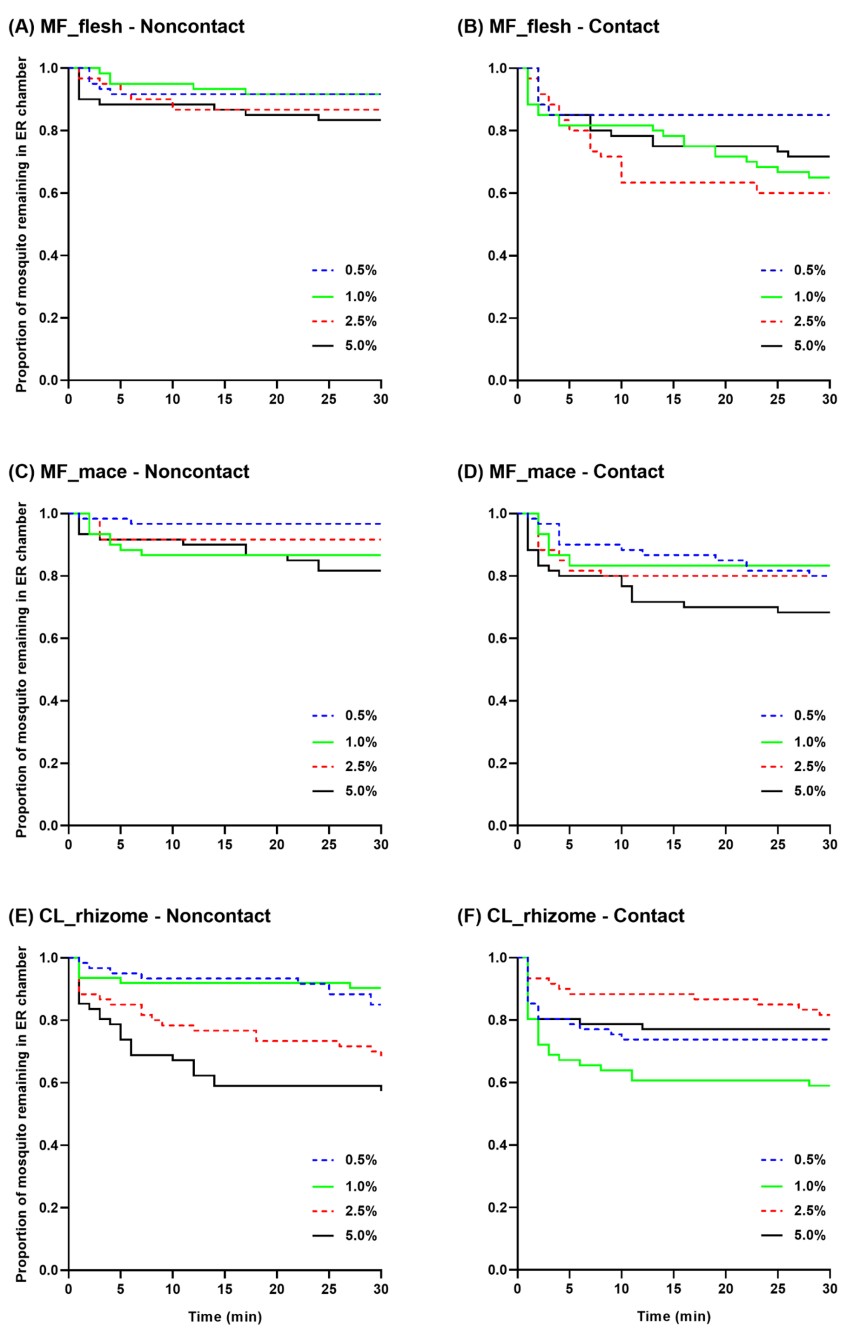

**Figure 4** **The proportion of mosquitoes remaining in the excito-repellency chamber for *Ae. aegypti* exposed to various concentrations of *Myristica fragrans* flesh (A, B) and mace (C, D) extract and *Curcuma longa* rhizome extract (E, F) in the treated non-contact and contact trials.** Escape responses were recorded at 1 min intervals during a 30 min exposure period. Paired control escape responses are not shown.

extract exhibited 1.75 times more ER action compared to *M. fragrans*. Our findings are in accordance with a previous study. Lotion containing a 10% essential oil made from

**Table 2  Log-rank comparison of escape responses of *Aedes aegypti* within plant extracts and concentrations between non-contact and contact ER assay configuration.**

| Concentration | *P-value* (non-contact *vs.* contact) | | |
|---|---|---|---|
| (% w/v) | *M. fragrans* (flesh) | *M. fragrans* (mace) | *C. longa* (rhizome) |
| 0.5 | 0.2524 | 0.0051[*] | 0.0954 |
| 1.0 | 0.0004[*] | 0.6040 | <0.0001[*] |
| 2.5 | 0.0012[*] | 0.0679 | 0.0946 |
| 5.0 | 0.1324 | 0.0765 | 0.0323[*] |

Notes.
*Significant difference $P < 0.05$.
% w/v means percent of weight (g) of plant extract in the total volume of solution.

**Table 3  Log-rank comparison of escape responses of *Aedes aegypti* within ER assay configuration between concentrations of plant extracts.**

| ER assay | Conc. | *P*-value | | |
|---|---|---|---|---|
| | (% w/v) | *M. fragrans* (flesh) | *M. fragrans* (mace) | *C. longa* (rhizome) |
| Non-contact | 0.5 *vs.* 1.0 | 0.9603 | 0.0499[*] | 0.4028 |
| | 0.5 *vs.* 2.5 | 0.4020 | 0.2491 | 0.0256[*] |
| | 0.5 *vs.* 5.0 | 0.1644 | 0.0096[*] | 0.0005[*] |
| | 1.0 *vs.* 2.5 | 0.3696 | 0.3823 | 0.0031[*] |
| | 1.0 *vs.* 5.0 | 0.1596 | 0.4764 | <0.0001[*] |
| | 2.5 *vs.* 5.0 | 0.6017 | 0.1123 | 0.1808 |
| Contact | 0.5 *vs.* 1.0 | 0.0135[*] | 0.7273 | 0.0939 |
| | 0.5 *vs.* 2.5 | 0.0045[*] | 0.9127 | 0.2619 |
| | 0.5 *vs.* 5.0 | 0.0761 | 0.1094 | 0.6980 |
| | 1.0 *vs.* 2.5 | 0.5481 | 0.6391 | 0.0046[*] |
| | 1.0 *vs.* 5.0 | 0.4904 | 0.0546 | 0.0414[*] |
| | 2.5 *vs.* 5.0 | 0.2144 | 0.1405 | 0.4896 |

Notes.
*Significant difference $P < 0.05$.
% w/v means percent of weight (g) of plant extract in the total volume of solution.

*C. longa* rhizomes prevented *Ae. aegypti* bite longer than the *M. fragrans* lotion when applied to forearms (*Tawatsin et al., 2006a*). It has been previously suggested that different behavioral responses in insects can be stimulated according to the function of concentration and/or molecule ratios, which lead to different neuron receptor activation patterns or odor maps (*Knaden et al., 2012*; *Deletre et al., 2016*). It is possible that the rhizome extract used in our study contained a combination of molecules in a ratio more favorable for provoking the escape response of *Ae. aegypti*. Further studies are needed to evaluate the chemical component of each extract, both individually and in combination, to establish the most effective repellent ratio. Different mosquito species have been observed to respond differently to certain plant-derived compounds (*Sathantriphop et al., 2014*). Both *M. fragrans* and *C. longa* lotions provided a much longer protection time when tested against *An. dirus* and *Cx. quinquefasciatus* (*Tawatsin et al., 2006a*) compared to this study. In another study, *C. longa* oil caused no repellency against *Ae. aegypti* but did cause repellency against *An. minimus* and *Cx. quinquefasciatus* (*Phasomkusolsil & Soonwera, 2010*). These

**Table 4** Primary chemical constituents of *Myristica fragrans* and *Curcuma longa* extracts analyzed by gas chromatography-mass spectrometry.

| Plant names | Part used | Primary chemical constituents (%)[*] |
|---|---|---|
| *M. fragrans* | flesh | diethyl malate (56.45), maleic anhydride (8.69), cis-vaccenic acid (3.33), palmitic acid (3.22), methoxyeugenol (3.14), citraconic anhydride (2.82), linoleic acid (1.98), 5-hydroxymethylfurfural (1.85), γ-sitosterol (1.69), 2-Propoxy-succinic acid, dimethyl ester (1.37) |
| | mace | elemicin (11.68), (1*S*,2*R*)-2-(4-allyl-2,6-dimethoxyphenoxy)-1-(3,4,5-trimethoxyphenyl)propan-1-ol-rel- (10.06), myristicin (9.76), (1*S*,2*R*)-2-(4-allyl-2,6-dimethoxyphenoxy)-1-(3,4-dimethoxyphenyl)propyl acetate (7.90), licarin A (7.54), licarin B (7.00), (1*S*,2*R*)-2-(4-allyl-2,6-dimethoxyphenoxy)-1-(3,4-dimethoxyphenyl)propan-1-ol (6.71), 1-phosphacyclopent-2-ene, 1,2,3-triphenyl-5-dimethylmethylene (4.63), 6-methoxyeugenyl isovalerate (4.21), (*S*)-5-allyl-1,3-dimethoxy-2-((1-(3,4,5-trimethoxyphenyl)propan-2-yl)oxy)benzene (3.63) |
| *C. longa* | rhizomes | ar-turmerone (24.56), β-turmerone (15.24), α-turmerone (10.52), zingiberene (6.09), sesquiphellandrene (4.93), α-curcumene (2.46), (*E*)-atlantone (2.45), benzenemethanol, 4-methyl- α-(1-methyl-2-propenyl)-, (*R* *,*R* *)- (2.38), (*Z*)-γ-atlantone (2.05), (6*R*,7*R*)-bisabolone (1.68) |

Notes.

[*]Values in parentheses represent relative amounts (% area) of each chemical constituent.

studies suggest that night-biting mosquito species are more likely to be repelled by both plants. Thus, it would be worthwhile to investigate the responses of *Anopheles* and *Culex* species to both the *M. fragrans* and *C. longa* extracts to further characterize their ER activities.

Different plant extracts have been shown to have different combinations of repellency and irritancy that involve different physiological mechanisms (*Deletre et al., 2013*). Our findings suggest that contact irritancy is the primary action of the *M. fragrans* flesh and mace extracts, regardless of concentration. One possible explanation for this is that the chemical components of the crude *M. fragrans* extract have low volatility, that is, their weak volatilization from the treated papers to the atmosphere inside the ER chamber requires close contact with the mosquito for action. The extract can then interact with the nervous receptors on the tarsal appendages of *Ae. aegypti*, leading to escape responses. For non-contact spatial repellency, it has been recognized that numerous odorant receptors (ORs) on mosquito antenna are essential in the detection of botanical compounds (*Norris & Coats, 2017*). We also hypothesized that the antenna of *Ae. aegypti* could be harboring different or lower ORs that are able to mediate *M. fragrans* extract, resulting in the limited repellency found in our study. Moreover, as previously stated, the concentration and molecule ratios of the *M. fragrans* extracts may not be sufficient to activate the ORs of *Ae. aegypti*. For *C. longa*, irritancy and repellency were the primary and secondary actions in the production of the *Ae. aegypti* escape response, respectively. Similar to *M. fragrans*, we hypothesized that contact with low concentrations of *C. longa* extract would irritate

mosquitoes, causing them to escape shortly after tarsal contact. It is possible that *Ae. aegypti* may have ORs that detect *C. longa* at larger concentrations (increased volatilization) or that the chemical constituents of the *C. longa* extract could be more effective at stimulating these ORs, resulting in repellency at high doses. To test these hypotheses, additional tests are required to determine whether each extract induces electroantennogram responses in the antenna of *Ae. aegypti* before any conclusions could be made. Overall, it should be noted that, since the ER of *M. fragrans* and *C. longa* is likely species-dependent, testing against other mosquito species may reveal contrast repellency and irritancy results.

The *M. fragrans* and *C. longa* crude extracts in this study were not toxic to *Ae. aegypti*, even at the highest concentration (5.0% w/v = 50,000 ppm), when exposed to treated-filter paper inside the ER chamber. It has been previously suggested that mosquitoes can avoid collectively lethal doses at exposures of only 30 min or less (if they escaped) in a larger space within the ER chamber (33.5 × 22.5 × 22.5 cm) (*Sukkanon et al., 2020*). One limitation of this study is that the ER chambers were made of window-less stainless steels; therefore, we were unable to observe and confirm mosquito movement inside each chamber during the exposure period. Several studies have shown the toxic action of both plants against mosquitoes. Using the topical application method, the seed oil of *M. fragrans* exhibited knockdown and adulticidal effects that caused temporary paralysis in *Ae. aegypti* females within 60 min post-exposure that led to death (*da Rocha Voris et al., 2018*). In another study, the ethanolic water mixture extract of *M. fragrans* seed also highly promoted *An. stephensi* female mortality when exposed to treated-filter paper (*Senthilkumar, Varma & Gurusubramanian, 2009*). For *C. longa*, vaporizing the rhizome oil through a mosquito mat machine efficiently caused mortality in female *Ae. aegypti*, *An. stephensi*, and *Cx. quinquefasciatus* (*Prajapati et al., 2005*). Additionally, the rhizome oil was found to be harmful to *Anopheles gambiae* Giles larvae (*Ajaiyeoba et al., 2008*). These experiments clearly establish the feasibility of using both *M. fragrans* and *C. longa* extracts as a green pesticide and repellent method for mosquito control.

In the present study, the chemical components of each crude extract were measured using GC-MS analysis. Our study is the first to report that diethyl malate, a malate ester, is the main chemical component found in the ethanolic crude extract of *M. fragrans* flesh (mesocarp). No insect repellent study has investigated diethyl malate. The three main compounds discovered in the mace crude extract employed in this study were three phenylpropanoids: elemicin; (1*S*,2*R*)-2-(4-allyl-2,6-dimethoxyphenoxy)-1-(3,4,5-trimethoxyphenyl)propan-1-ol-rel-; and myristicin. These substances have also been reported in earlier research (*Periasamy et al., 2016*; *Kuete, 2017*; *Ha et al., 2020*). Although the pure compound of elemicin has never been studied for its insect repellency, it has been found that the essential oils of *Cymbopogon* grasses, which contain a high level of elemicin (8%–56%), provided great protection against *Ae. aegypti*, *An. stephensi*, and *Cx. quinquefasciatus* bites (*Tyagi, Shahi & Kaul, 1998*). To our knowledge, the mosquito repellent effectiveness of myristicin has not been demonstrated; however, myristicin has shown potent larvicidal activity against *Culex pipiens pallens* L. by lowing water surface tension (*Bae et al., 2017*). Due to the lack of attention from the repellency standpoint, it would be of great interest to investigate the role of these components (*e.g.*, diethyl malate,

elemicin, myristicin) against mosquito vectors. Our study also found ar-turmerone, β-turmerone, and α-turmerone in the crude *C. longa* rhizome extract, whose effectiveness has similarly been demonstrated in previous studies (*Das, 2016*; *Lim, 2016*). Ar-turmerone showed higher biting deterrent activity against *Ae. aegypti* when compared to DEET, but its activity against *Anopheles quadrimaculatus* Say was comparable to DEET (*Ali, Wang & Khan, 2015*). Thus, it is possible that ar-turmerone alone or in combination with other compounds could be responsible for the escape response of *Ae. aegypti* observed in our ER study.

The enhancement in repellent efficiency of essential oils using synergistic phenomena has been observed in previous studies. For example, when testing against *Ae. albopictus* in both laboratory and field trials, the repellency of the 1:1:2 mixture of the *C. longa* rhizome, *Zanthoxylum limonella* (Dennst.) Alston. fruit, and *Pogostemon heyneanus* Benth. leaf oils was greatly improved compared to each individual oil (*Das et al., 2015*). Another interesting repellency improvement is the addition of fixative materials, such as vanillin. The addition of 5% vanillin significantly improved the repellency of 10% *C. longa* rhizome oil from 66 min protection time to 120 min (82% change) against *Ae. aegypti* using the arm-in-cage assay (*Auysawasdi et al., 2016*). Further studies could be performed by creating mixtures of *M. fragrans* and *C. longa* extracts with other plant extracts or oils with or without the addition of fixative agents (*e.g.*, vanilin) to improve their ER actions against mosquito vectors.

## CONCLUSIONS

In the present study, the *M. fragrans* flesh and mace extracts above 1.0% and 2.5% concentrations, respectively, produced the greatest ER properties, predominantly in contact irritancy. The highest contact irritancy in the *C. longa* rhizome extract was initially found at a concentration of 1.0%, followed by non-contact repellency at 5.0%. Based on our hypotheses and findings, the crude *M. fragrans* and *C. longa* extracts elicited mosquito avoidance behavior as contact irritants against *Ae. aegypti*, a vector of the Zika virus. Despite the limited ER observed, other actions of both plants, such as larvicidal and adulticidal activities, have been demonstrated in other studies. Thus, more comprehensive studies of these two plants are needed to develop them as a key alternative in *Aedes* control. Further investigations are also required to illustrate the active ingredients of both *M. fragrans* and *C. longa* extracts and oils, which may be favorable for the development of green mosquito repellent products. Testing against other mosquito species, particularly night-biting mosquitoes, and the varied insecticide-susceptible field mosquito populations, are also vital to fully characterize the ER of both plants. Local residents would benefit from using plant materials cultivated in their community as a source of repellent development. More experiments must be performed to enhance the ER efficacy of these plants, to formulate a practical product, and to provide user-friendly access.

## ACKNOWLEDGEMENTS

We are grateful to the Department of Entomology, Faculty of Agriculture, Kasetsart University for providing laboratory space and equipment. We thank Dr. Jirod Nararak and Miss Pairpailin Jhaiaun, Department of Entomology, Faculty of Agriculture, Kasetsart University, for assisting during the entire experimental period. We thank Mr. Vasu Krittiyamethakran, Department of Medical Technology, School of Allied Health Sciences, Walailak University, for assisting in illustration design. The authors would like to thank Enago for the English language review.

### Funding

This work was supported by the Research Institute for Health Sciences, Walailak University (Grant No. WU-IRG-64-014) and the Kasetsart University Research and Development Institute (KURDI) (Grant No. FF (KU) 14.64). The funders had no role in study design, data collection and analysis, decision to publish, or preparation of the manuscript.

### Grant Disclosures

The following grant information was disclosed by the authors:
The Research Institute for Health Sciences, Walailak University: WU-IRG-64-014.
The Kasetsart University Research and Development Institute (KURDI): FF (KU) 14.64.

### Competing Interests

The authors declare there are no competing interests.

### Author Contributions

- Phuangthip Bhoopong and Chutipong Sukkanon conceived and designed the experiments, performed the experiments, analyzed the data, prepared figures and/or tables, authored or reviewed drafts of the paper, and approved the final draft.
- Theeraphap Chareonviriyaphap conceived and designed the experiments, authored or reviewed drafts of the paper, and approved the final draft.

### Ethics

The following information was supplied relating to ethical approvals (i.e., approving body and any reference numbers):

The protocol used for animals was approved by Walailak University Animal Ethics Committee (Reference No. WU-ACUC-64034).

### Data Availability

The raw escape data (number of escaped mosquitoes) of *Aedes aegypti* at 1 min interval during 30 min exposure to various concentrations of plant extracts inside both non-contact and contact chambers of the excito-repellency assay system are available in the Supplementary Files.

## Supplemental Information

Supplemental information for this article can be found online at http://dx.doi.org/10.7717/peerj.13357#supplemental-information.

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
