# Peer review of "Excito-repellency of Myristica fragrans Houtt. and Curcuma longa L. extracts from Southern Thailand against Aedes aegypti (L.)"

_PeerJ, doi:10.7717/peerj.13357_

## Round 0.1 · original submission · Major Revisions

Dear Authors

Thank you for you submission in the PeerJ. The overall attitude of the reviewers is good but have some major/minor changes. Please incorporate according to the attached comments.

Reviewer 1 ·

Basic reporting

Dear Authors
Overall study is scientifically sound but has shortcoming to incorporate before consideration.
About title
I think that this is unsuitable to state here because study id repelling the A. egypti....... Please change this in the title replacing vector with mosquito name.......
Introduction
Please make the objective more clear by adding the data of repellent background in the last paragraph.
please add the more data of the biochemical potential of plant extracts..... which will be actual background of the study...
Line 49-51: Please make two sentences and rewrite it.
Line 118-119: Move in the materials and methods.
Materials and Methods
Line 123: change was with were
In the discussion the comparison of previous studies is not bad but it can be accepted somehow...... Authors almost have more than half discussion of this nature which i do not think a good in publication...... Please make the discussion good according to the trends of your results....... Discuss your plants extract potential/efficacy rather than discussing other plants......
Line 279-280: Above in introduction you have mentioned that this extracts has never been tested before?????
In the conclusion, Please conclude some results.

Experimental design

Overall Methods of study is explained in considerable but some queries are still to answer
Why you did not use the mean comparison tests (Tuckeys, Dunnetts etc.), while you have different treatments groups.....
I do not think that without theses authors can compare which is better one.......
Where is replication scheme???
if pseudo replication then how authors manage the experimentation????
Roberts et al. 1997 methods: Please explain here and cite at the end of text........

Validity of the findings

Overall the results are satisfactory....
On more analysis must be in the results section......
mean comparison test using tuckeys test.....

Annotated reviews are not available for download in order to protect the identity of reviewers who chose to remain anonymous.

Reviewer 2 ·

Basic reporting

Overall study of this paper is good but I am concerned that the paper should be improved to meet the standard of PeerJ for publication. The manuscript in its current form is difficult to considered for publication, as it need to resolve suggested changes.
Title is not adequate for the content and given information of this paper. It must be precise and clear according to the data mentioned in methodology.
Abstract is written well but need addition of future directions.
Introduction data is not sufficient to discuss the background of the designed study. Please add more relevant text to improve the reading context of the title studied. Remove old references. The hypothesis should be clearly stated in the introduction, which can never be modulated with a strong story background of the study. It is suggested to improve it.
L 80-82: Need to rephrase this sentence and citations must be at the end of this sentence.
L 83-86: Citation at the end of sentence.
L 97: DEET, explain this word
L 118 and 119: No need to mention figure here.
The authors need to precise the data in discussion and remove old references.

Experimental design

The experimental design of this study is well explained. Need few changes i.e.,
L 183: 24-h “24 h”
L 185: 3-min “3 min”
L 189: 30-min “30 min”

Validity of the findings

The results are good for the presentation of this study.

Additional comments

I would like to commend the authors for their efforts, but there is need to make the paper interesting for an international audience such as the PeerJ readers.

Reviewer 3 ·

Basic reporting

It is an interesting study having a sufficient research data and well explained results in an effective way. More interestingly, it is the first study in this field that illustrate the effect of two plant materials on the zika virus. Here are few minor changes in the research article Please rectify all with keen concern to improve the paper.
It seems there is need to rectify the font size of all headings, also make it sure that all headings and sub-headings have same font. “Justify” all paragraphs in the manuscript. Furthermore, many grammatical errors have been observed in the manuscript, please read it again carefully and remove all obstacles. Sentence making also need to improve to clarify the central idea of the research, as at some points in the paper it creates ambiguity in the actual meaning of sentence due to poor sentence making. Objectives of this research are unclear, Hence, there is need to more explanation in comprehensive way of research objectives in the end of “Introduction”. Figures are unnecessary in the introduction, please add research relevant plants figures in the materials and methods if seems necessary. Few suggestions are following:
1- line 47 of introduction replace mosquitos to mosquito
2- line 48 also give other examples of “several tropical disease pathogens”
3- Line 51, Remove “Is Concerning”
4- Line 52 Remove “At least”
5- Line 53 Replace is >Was
6- Line 56 Replace till date>Until now
7- Line 57 Replace To be >is
8- Line 65 Remove “and” before readily available
9- Line 67 Remove “Since”
10- Line 69 replace Can be >Could be
11- Line 73, 74. There is no need for any figure in introduction. You might include it in the Materials and Methods if it would be the part of your Research work.
12- Line 76, remove “Symptoms”
13- Line 90, No need for any figure in introduction
14- Line 98 Replace Rhizome essential oils>its essential oil
15- Line 118 No need of figures in introduction
16- Line 118-119 seems unfit in introduction, must be the part of materials and methods.

Experimental design

This part is very well explained with adequate detail of each step of the experiment. Meanwhile, few minor points need to edit to improve the significance of paper. Please add references of all protocols used in this research. “Data analysis” part of this section should be at the end of materials and methods. It would be more beneficial for better understanding if number of plants also mentioned in this section. Figures of plants and extracts used in research material should add in this part. Few more points are following:
1- Line 142 remove “and” before mace
2- Line 166 remove “or”
3- Line 179 replace are >were
4- Line 180 replace is >was
5- Line 187 replace was>were
6- Line 191 replace was>were
7- Line 196 to 206 must be the last lines of Materials and Methods. For this purpose shift the complete heading data of “Data analysis” after the “Gas chromatography-mass spectrometry (GC-MS) analysis”.

Validity of the findings

Research results are well explained, and novelty of the research is well describe in this section in a brief way to meet all requirements of research objectives. Albeit on, this section needs more concentration on correct sentence making and improvement in grammar as well. Please use less quantitative values to support the results in discussion, it creates cliché. Some points need more justification to support the results
1- Lines 250-257 required more justification/references to support the results
2- Line 334 needed more references to support the results

Additional comments

I have keenly review the manuscript and try my best to fulfil the general requirements of the journal. It is an interesting study having all the essential requirements of a good paper especially, the authors have tried to manipulate the new research area in best possible way.

---

## Round 0.2 · accepted · Accept

Dear Authors

Thank you for choosing PeerJ for publication. Based on the reviewers' reports I accept the submission but a few minor changes are needed in the final version.

Comments are attached herewith.

Reviewer 1 ·

Basic reporting

I suggest the the title below because it is still not looks very impressive
"Excito-repellency of Myristica fragrans Houtt. and Curcuma longa L. extracts against Aedes aegypti (L.) in Southern Thailand"
You may choose any other on your choice.

Experimental design

NA

Validity of the findings

NA

Additional comments

NA

Reviewer 2 ·

Basic reporting

The authors resolved all the suggested changes that I have mentioned previously. In present form the paper is according to the standard of PeerJ for publication. I appreciate to the authors that they did a lot of hard work for this study.

Experimental design

N/A

Validity of the findings

N/A

Additional comments

N/A

Reviewer 3 ·

Basic reporting

This study is scientifically clear and advanced knowledge have been given in the article. The background knowledge is also sufficient and no ambiguity found in the data. Results are according to the hypothesis and fulfil the requirements of the study. Author has provided enough data to publish this manuscript in the PeerJ.
just Minor change is recommended:

L96: Replace “Curcuma longa”> C. longa
L122-123: Remove “To the best of our knowledge, the ER properties of M. fragrans 123 and C. longa extracts have never been accurately studied” as it is repeated in next paragraph with more detail.

Experimental design

Research question is well defined and relevant to the study. Methodology is also clearly explained with sufficient information about the research.

Validity of the findings

All provided data is statistically sound and robust. conclusions is also well stated and linked to the primary research objective.

Additional comments

N/O